# Multiscale Entropy Quantifies the Differential Effect of the Medium Embodiment on Older Adults Prefrontal Cortex during the Story Comprehension: A Comparative Analysis

**DOI:** 10.3390/e21020199

**Published:** 2019-02-19

**Authors:** Soheil Keshmiri, Hidenobu Sumioka, Ryuji Yamazaki, Hiroshi Ishiguro

**Affiliations:** 1Hiroshi Ishiguro Laboratories (HIL), Advanced Telecommunications Research Institute International (ATR), 2-2 Hikaridai Seika-cho, Kyoto 619-02, Japan; 2School of Social Sciences, Waseda University, 1 Chome-104 Totsukamachi, Shinjuku, Tokyo 169-8050, Japan; 3Graduate School of Engineering Science, Osaka University, 2-1 Yamadaoka, Suita, Osaka 565-0871, Japan

**Keywords:** multiscale entropy, embodied media, tele-communication, humanoid, prefrontal cortex

## Abstract

Todays’ communication media virtually impact and transform every aspect of our daily communication and yet the extent of their embodiment on our brain is unexplored. The study of this topic becomes more crucial, considering the rapid advances in such fields as socially assistive robotics that envision the use of intelligent and interactive media for providing assistance through social means. In this article, we utilize the multiscale entropy (MSE) to investigate the effect of the physical embodiment on the older people’s prefrontal cortex (PFC) activity while listening to stories. We provide evidence that physical embodiment induces a significant increase in MSE of the older people’s PFC activity and that such a shift in the dynamics of their PFC activation significantly reflects their perceived feeling of fatigue. Our results benefit researchers in age-related cognitive function and rehabilitation who seek for the adaptation of these media in robot-assistive cognitive training of the older people. In addition, they offer a complementary information to the field of human-robot interaction via providing evidence that the use of MSE can enable the interactive learning algorithms to utilize the brain’s activation patterns as feedbacks for improving their level of interactivity, thereby forming a stepping stone for rich and usable human mental model.

## 1. Introduction

Socially assistive robotics (SAR) [1] is an emerging field of research that focuses on intelligent and interactive media to provide assistance through social than physical means [2]. SAR builds upon the behavioral and neuroscientific findings of the positive motivational effect of physically embodied media on humans’ social inclinations [3,4]. For instance, it identifies that children who read with a learning companion robot consider their reading companion to support their reading comprehension and that it motivates a deepening social connection [5]. Moreover, it indicates that tele-communication through a humanoid results in the older people’s brain to exhibit a similar activation pattern as in the case of in-person communication [6].

In this regard, a distinct attribute of robotic media is their physical embodiment which allows for a sense of togetherness [7]. This property can amplify human work [1] by, for example, filling the gap in human personnel shortage in elderly care facilities [8,9]. Such potentials become more intriguing by considering the positive effect of these media in robot-assistive cognitive training of the older people [10].

On the other hand, communication research suggests [11] the tendency of individuals to undermine the significant influence of the physical embodiment during verbal communication in a wide range of social [12] and behavioral domains [13]. However, almost all of these findings are based on the subjective assessments of the behavioral responses of the human subjects. This severely limits the possibility of drawing a reliable conclusion. In fact, there is a paucity of research on the neurophysiological effect of physical embodiment in communication research.

The human brain, as with any other healthy physiological system, is an inherently complex system whose dynamics strongly correlate with the productivity of its cognitive functions such as attention and language [14]. Interestingly, an increase in complexity reflects the information content of the dynamics of physiological systems, which, in turn, have a direct correspondence to the variational information in their activities [15]. Therefore, the presence or the absence of a potential shift in the dynamics of the brain activity in response to a physically embodied medium can provide evidence on the (lack of) significance of the physical embodiment during verbal communication.

In this article, we argue that if the medium reinforces the perception of the conveyed message [16] then the embodiment is inevitably an essential part of the communicated content. To verify our claim, we performed a multiscale entropy (MSE) analysis [17,18,19,20] of the older people’s prefrontal cortex (PFC) responses to tele-communicated stories, in which we communicated these stories with the older people through a speaker, a video-chat system, and a humanoid. In addition, we considered face-to-face to be a control setting, through which we communicated these stories with participants in-person. We chose MSE due to its discriminative power in detecting the change in complexity of biological signals [21,22]. Furthermore, we considered older participants due to the research findings that indicate the reduction of neurophysiological signal complexity by aging [23,24,25,26,27]. This allowed us to utilize MSE as a reliable biomarker for detection of any potential increase in the dynamics of the older people’s brain activity in response to physical embodiment. We chose storytelling since stories’ scripts can be kept intact and repeated to different individuals without any change in their contents, thereby allowing for the control of such confounders as subtle differences in conveyed information. We chose PFC due to neuroscientific findings on its pivotal role in language processing [28], social cognition [29], and story comprehension [30,31,32]. Our objective was to verify the following hypothesis.

**Hypothesis** **1.***Embodied media differentially stimulate the dynamics of the older people’s PFC during a tele-communicated verbal communication*.

Our contributions are threefold. First, we provide evidence that physical embodiment induces a significant increase in MSE of the older people’s PFC activity. Second, we show that such a shift in dynamics of the older people’s PFC activity significantly reflects their perceived feeling of fatigue that is induced by listening to the stories. Third, we show that the increase in MSE by physical embodiment significantly differentiates the individuals whose perceived feeling of fatigue are above its average perception by the older people population in our study.

Our results benefit the researchers in age-related cognitive function and rehabilitation [23] who seek for the adaptation of these media in robot-assistive cognitive training of the older people [10]. In addition, they offer a complementary information to researchers in the field of human–robot interaction for modeling the human behavior. For instance, robotic media with the ability to detect the perceived feeling of fatigue by their human companions can close the gap on attaining a sustained verbal communication with them via adapting to individuals’ pace and interest in response to conversational nuances and complexity. Moreover, changes in the pattern of brain activation (as reflected by the MSE) can enable the interactive learning algorithms [33] to utilize the brain’s activation patterns as feedbacks for improving their level of interactivity. This, in turn, can form a stepping stone for rich and usable model of human mental state [34]. To the best of our knowledge, this is the first study that utilized MSE to investige the effect of physical embodiment on human subjects’ brain activity.

## 2. Results

Figure 1A shows the grand-average MSEs of the older people’s left-hemispheric PFC in different media settings. In this plot, scale factors 10 and 20 correspond to the one-second and two-second data acquisition intervals, given the sampling rate of our device (i.e., 10.0 Hz). We observed an increase in MSEs between the first two scale factors that was followed by a reduced MSEs up until scale factor 8. In addition, these MSEs exhibited a subtle increase at around one second data acquisition. Moreover, they roughly followed a straight line passed the 10 scale factor. We also observed that the MSEs of the older people’s PFC was, on average, highest in the case of physically embodied medium (magenta), followed by the in-person setting (red).

Friedman test identified a significant effect of media on older people’s MSEs in different media settings (*p* < 0.001, H(3, 79) = 51.12, r= 0.80). Post hoc Wilcoxon test identified that face-to-face setting induced a significant increase in MSEs of older people’s left-hemispheric PFC in comparison with speaker (*p* < 0.05, W(38) = 2.12, r= 0.34, M${}_{F}$ = 0.67, SD${}_{F}$ = 0.10, M${}_{S}$ = 0.62, SD${}_{S}$ = 0.11) and video-chat (*p* < 0.001, W(38) = 3.69, r= 0.58, M${}_{V}$ = 0.57, SD${}_{V}$ = 0.09). Similarly, Telenoid induced an increase in MSE values that was significantly higher than speaker (*p* < 0.01, W(38) = 2.88, r= 0.46, M${}_{T}$ = 0.69, SD${}_{T}$ = 0.11) and video-chat (*p* < 0.001, W(38) = 3.61, r= 0.57). Moreover, speaker induced a significantly higher MSE than video-chat (*p* < 0.05, W(38) = 2.15, r= 0.34). On the other hand, we observed a non-significant difference between Telenoid and face-to-face (*p* = 0.063, W(38) = 1.85, r= 0.29). Figure 1B shows these results.

Whereas we observed no correlation between older people’s MSEs and their self-assessed responses to the feeling of fatigue in speaker (Figure 2S) (r=−0.05, *p* = 0.85, M${}_{Fatigue}$ = 3.67, SD${}_{Fatigue}$ = 1.80), video-chat (Figure 2V) (r=−0.13, *p* = 0.65, M${}_{Fatigue}$ = 4.20, SD${}_{Fatigue}$ = 2.11), and face-to-face (Figure 2F) (r=−0.16, *p* = 0.57, M${}_{Fatigue}$ = 3.80, SD${}_{Fatigue}$ = 2.21), we found a significant anti-correlation in the case of Telenoid (Figure 2T) (r=−0.57, *p* < 0.03, M${}_{Fatigue}$ = 4.33, SD${}_{Fatigue}$ = 2.12).

### Prediction of the Older People Perceived Fatigue Using MSE

We observed a significant anti-correlation between MSEs of the older people’s left hemisphere and their self-assessed responses to the feeling of fatigue in the Telenoid setting. This suggested the potential utility of MSE for predicting the perceived level of fatigue by older people based on MSE of their frontal brain activity. To investigate this possibility, we interpreted the mean of the MSE clusters in different media settings as their respective decision boundaries. Figure 3 shows the MSE clusters associated with the older people’s left PFC. Clusters’ boundaries are depicted in black line segments in this figure.

Table 1 shows the metrics associated with the use of left-hemispheric MSEs of the older people for prediction of their perceived the feeling of fatigue. Significantly above chance prediction accuracy of the MSEs in the case of the Telenoid is evident in this table.

Figure 4 presents the confusion matrix pertinent to the prediction of the perceived feeling of fatigue using the left-hemispheric MSEs. We observed that the use of the MSEs achieved the highest TP and TN in case of the Telenoid. Similarly, FP and FN were smallest in the case of the Telenoid in comparison with other media settings. On other hand, speaker, video-chat, and face-to-face media settings achieved a comparable FP and FN. These results were in accord with the correlation analysis of the MSEs of the older people’s left PFC.

## 3. Discussion

We argued that the physical embodiment is inevitably an essential part of the communicated content and therefore the embodied media differentially stimulate the dynamics of the individuals’ PFC during a tele-communicated verbal communication. We verified our claim through MSE analysis [17,18] of the older people’s PFC responses to tele-communicated stories in which we communicated these stories with the older people through a speaker, a video-chat system, and a humanoid. In addition, we considered face-to-face to be a control setting, through which we communicated these stories with participants in-person.

We found evidence in support of our hypothesis. Precisely, the analysis of the older people’s MSE identified that the physically embodied medium and the in-person setting induced an increase in the dynamics of their PFC activity. This increase that was significantly different from the speaker and the video-chat settings was non-significant between the physically embodied medium and the in-person setting. We also observed that the physically embodied medium and the in-person setting induced a bilateral PFC activation. This observation was in line with the neuroscientific findings on bilateral effect of stories on human subjects’ PFC [30,31,32]. It also complemented these previous results by identifying that such an effect was not only present in the PFC’s dynamical complexity but also the embodiment of the medium through which such contents were communicated had significant impact on the induced PFC activity.

We also observed that the older people’s MSEs were higher in the finer time scales, which was accompanied with their decrease in the coarser scales. This was in accord with the findings that attribute such an effect to a shift from long-range brain regions connectivity (captured by the coarse scale MSEs) to a more local processing by aging [22,25]. Moreover, these MSEs tended to straight lines from scale factors 12 to 20. Such a tendency that is a characteristic of nonlinear systems [35,36] suggested the existence of a power law property [24]: an intrinsic complexity criterion in physiological systems [37]. In this regard, the ability of the physically embodied medium (similar to the in-person setting) to increase the complexity of the older people’s brain activity implied the potential utility of embodiment in healthy cognitive aging through consideration of the correlation between increased physiological complexity and healthy condition [14,37].

Our further analyses identified that the shift in dynamics of the older people’s PFC activity by physically embodied medium reflected their perceived feeling of fatigue and that such a shift was sufficient to significantly differentiate the individuals whose perceived feeling of fatigue were above the average perception of the older people population in our study. Additionally, they implied that such a differentiability was significantly stronger in the older people’s left- than right-hemispheric PFC. Surprisingly, we found no correlation between the older people’s MSEs and their perceived feeling of fatigue in the in-person setting, which was also in accord with the significantly low prediction accuracy in this condition. This observation appeared to undermine the effect of physical embodiment through such counterarguments as since the presence of a human is the best embodied representation for a human, the lack of correlation in this case may also suggest the observed correlation in the case of physically embodied medium to be spurious. However, several scientific studies identify the utility of tele-communication to surpass the in-person setting [38,39,40,41]. For instance, Joinson [42] noted the effect of the former in increasing the willingness of the individuals for self-disclosure in contrast with the in-person communication. Along the same direction, Zimmerman [43] concluded that, in contrast with the in-person communication, such settings as computer-mediated communication (CMC) [44] elicit emotionally rich, relationship-oriented verbal interaction among emotionally disturbed adolescents. In light of these findings, the observed phenomenon in our results may point at the utility of physically embodied medium in moderating the stress and unwillingness of individuals during their verbal communication with an unknown person. Although this interpretation finds evidence in the non-significant difference of the older people’s increase in MSE in the case of physically embodied medium and in-person setting, further research is necessary to draw a more informed conclusion.

Our results contribute to such socially assistive robotics [1,2] scenarios as child education and elderly care. For instance, they suggest that the use of MSE can enable these media to determine whether their level of interaction (e.g., socialization [45], reading and comprehension [5]) is exceeding the comfort level of children, thereby allowing for modulation of their communicated contents and/or behavioral interaction. Similarly, they can advance the use of these media in elderly care facilities by enabling them to act as cognitive training mediators with exclusive access to older people’s PFC dynamical changes during their cognitive training to determine their comfort in continuing their cognitive task [8,9]. The latter potential becomes more intriguing, considering the positive effect of these media in robot-assistive cognitive training of the older people [10].

Our results show a promising first step toward the use of brain information for quantification of one of the basic component of the human mental state: perceived fatigue due to the verbally communicated contents. These results benefit the researchers in age-related cognitive function and rehabilitation [23] that seek for adaptation of these media in robot-assistive cognitive training of the older people [10]. They also benefit the human–robot interaction research through such paradigms as interactive learning [33] in which such algorithms can utilize the brain’s dynamical changes in the form of MSE as feedbacks for improving their level of interactivity. This, in turn, can form a stepping stone for rich and usable models of human mental state [34]. In a broader perspective, the ability to estimate the perceived feeling of fatigue during a humanoid-mediated verbal communication can contribute to the study and analysis of a robotic theory of mind (ToM) [46] through critical investigation of its implications in humans’ neurological responses while interacting with their synthetic companions.

### Limitations and Future Direction

Although our results indicated the significant role of the physical embodiment during verbal communication, a larger human sample is necessary for an informed conclusion on the utility of these results. Moreover, our participants were limited to older people. Therefore, it is necessary to investigate the effect of the physical embodiment in other age groups (e.g., kids, adolescents, and younger adults) to verify that our results are not affected by this factor.

Considering the utility of the physically embodied medium in differentiating the older people’s perceived feeling of fatigue in conjunction with the absence of such a differentiability in the case of face-to-face setting, it is adequate to question the potential role of the novelty effect on this result. Therefore, further investigation in a longitudinal setting in which older people participate in multiple conversational sessions is necessary to verify that our findings are not affected by the long-term exposure to such media.

Moreover, our experimental setting was limited to a storytelling in which participants listened to a verbally communicated content without any requirement for their response. Therefore, it is crucial to analyze the effect of the physical embodiment in the conversational scenarios to examine the effect of such bidirectional verbal interaction on dynamical changes of the PFC.

In addition, the present study did not include other types of physically embodied media (e.g., mechanical looking robots, pet robots, etc.). Therefore, it is necessary to determine the correspondence between the media embodiment and the PFC’s dynamics. It is also important to verify whether different embodiments can induce differential impact on the prediction accuracy of the feeling of fatigue.

## 4. Materials and Methods

### 4.1. Participants

Our participants consisted of twenty older people (ten females and ten males, 62–80 years old, MEAN = 70.70, SD = 4.62). All participants were free of neurological or psychiatric disorders and had no history of hearing impairment. Since we failed to collect the NIRS time series of the brain activity of four elderly adults due to recording complications, they were excluded from our analyses. We used an employment service center for older people to recruit our participants. This study was carried out in accordance with the recommendations of the ethical committee of Advanced Telecommunications Research Institute International (ATR) with written informed consent from all subjects. The protocol was approved by the ATR ethical committee (approval code:16-601-1).

### 4.2. Communication Media

Our experimental settings included a humanoid robotic medium (Figure 5a), an audio speaker (Figure 5c), a video-chat system (Figure 5d), and a human (i.e., in-person, Figure 5e). We chose a minimalist teleoperated android called, Telenoid R4^TM^ (Telenoid hereafter, Figure 5b) [47]. Telenoid is approximately 50.0 cm long and weighs about 3.0 kg. It comes with nine degrees-of-freedom (3 for its eyes, 1 for its mouth, 3 for its neck, and 2 for its arms) and is equipped with an audio speaker on its chest. It is primarily designed to investigate the basic and essential elements of embodiment for the efficient representation and transfer of a humanlike presence. Therefore, its design follows a minimalist anthropomorphic principle to convey a gender-and-age-neutral look-and-feel. In the present study, we chose a minimalist anthropomorphic embodiment to eliminate the projection of such physical traits as gender and age onto our robotic medium.

Telenoid conveyed the vocal information of its teleoperator through its speaker. Its motion was generated based on the operator’s voice, using an online speech-driven head motion system [48]. However, its eyes and arms were motionless in this study. We placed Telenoid on a stand approximately 1.40 m (Figure 5a) from the participant’s chair to prevent any confounding effect due to tactile interaction (e.g., holding, hugging, etc.). We adjusted this stand to resemble an eye-contact setting between Telenoid and the participant. We maintained the same distance in the case of the other media as well as for the in-person setting. In the in-person condition (Figure 5e), we adjusted the storyteller’s seat to maintain eye-contact with the participant. For the video-chat (Figure 5d), we adjusted its placeholder in such a way that the storyteller’s appearance on the screen resembled an eye-contact setting. In the speaker setting (Figure 5c), we placed the video-chat screen in front of the participant (as in the video-chat condition) and placed the speaker behind its screen.

We used the same audio device in the speaker and video-chat settings to prevent any confounding effect due to audio quality. We used the same recorded voice of a woman, who was naive to the purpose of this study, in speaker, video-chat, and Telenoid. These recordings took place in a single session in which we recorded her voice and video while telling stories. In Telenoid setting, we played back the same prerecorded voice for the speaker through the audio speaker on its chest. In the in-person setting, the same woman read the stories to the participants.

We asked our female storyteller to stay as neutral as possible while reading these stories. However, we are unable to confirm the absence of any difference in emotional impact of the stories’ content on her during the in-person or the voice/video recordings.

### 4.3. Sensory Device

We used functional near infrared spectroscopy (fNIRS) to collect the frontal brain activity of the participants and acquired their NIRS time series data using a wearable optical topography system called “HOT-1000”, developed by Hitachi High-Technologies Corp. (Figure 6). Participants wore this device on their forehead to record their frontal brain activity through detection of the total blood flow by emitting a wavelength laser light (810 nm) at a 10.0 Hz sampling rate. Data acquisition was carried out through four channels (L1, L3, R1, and R3, Figure 6). Postfix numerical values that are assigned to these channels specify their respective source-detector distances. In other words, L1 and R1 have a 1.0 cm source-detector distance and L3 and R3 have a 3.0 cm source-detector distance. Note that, whereas a short-detector distance of 1.0 cm is inadequate for the data acquisition of cortical brain activity (e.g., 0.5 cm [49], 1.0 cm [50], 1.5 cm [51], and 2.0 cm [52]), 3.0 cm is suitable [49,51].

Findings on the brain activation during memory and language processing suggest a left-lateralized activation in both genders with higher specificity in females [53,54]. Therefore, we reported the results pertinent to $Left_{3}$ in the main body of our article (Appendix A for results on $Right_{3}$).

### 4.4. Paradigms

Our experimental paradigm consisted of storytelling sessions in which a woman narrated three-minute stories from Greek mythology in the selected media settings. This resulted in four separate sessions per participant. To prevent any confounding effect of visual distraction and to control the field of view of the participants within the same spatial limit, we placed their seat in a cubicle throughout the experimental sessions (height = 130.0 cm, width = 173.0 cm, depth = 210.0 cm). This cubicle’s side wall is visible in blue in Figure 5a–e.

Every participant first gave written informed consent in the waiting room next to the experimental room. Then, a male experimenter explained the experiment’s full procedure to the participants. This included the total number of session (four sessions), the duration of the narrated story in each session (i.e., three minutes), instructions about the one-minute rest period prior to the actual session (i.e., sitting still with eyes closed), and the content of the stories. The experimenter also asked the participant to focus on listening to the stories that were either narrated by a woman through a medium or in-person. Next, he led the participants to the experimental room and helped them to get seated in an armchair with proper head support in a sound-attenuated testing chamber. Then, the experimenter calibrated the eye tracker device and instructed the participants to fully relax and keep their eyes closed. Last, the experimenter verified the proper adjustment of the medium (or helped the storyteller get comfortable in the proper position for the in-person condition) and began the experimental session. In every session, we first acquired one-minute rest data. Next, the experimenter asked the participants to open their eyes and prepare to listen to the story, followed by a story session during which we recorded the NIRS time series of the frontal brain activity of the participants. During the in-person setting, we asked the storyteller to maintain as much eye-contacts with the participants as possible.

At the end of each session, all participants completed a questionnaire on their perceived (on an 8-point scale with 1 = not at all and 8 = very much) of feeling of fatigue due to listening to the story. We provided our participants a one-minute rest period prior to the commencement of each of the storytelling sessions and asked them to keep their eyes closed. In this period, we prepared the setting for the next storytelling session. We video-recorded all the activities throughout the experiment.

Every subject participated in all four sessions. We kept the content of the stories intact in these sessions and randomized the order of the media among the participants without changing the order of the narrated stories. The entire experiment lasted about 90 minutes per participant.

### 4.5. Data Processing

To attenuate the effect of systemic physiological artefacts [55] (e.g., cardiac pulsations, respiration, etc.), we applied a one-degree polynomial Butterworth filter with 0.01 and 0.6 Hz for low and high bandpass which was then followed by performing the linear detrending on the data. Detrending of the signal that was adapted from signal processing and time series analysis and forecasting was a necessary step to ensure that the assumptions of stationarity and homoscedasticity (as reflected in wide spread application of linear models in analysis of fNIRS/fMRI time series) were not strongly violated (e.g., due to seasonality and/or repetitive increasing/decreasing patterns). Finally, we attenuated the effect of the skin blood flow (SBF) using an eigen decomposition technique [56]. This approach considers the first three principal components of all NIRS recorded channels of the participants’ frontal brain activity during rest period to represent the SBF. Subsequently, it eliminates the SBF effect by removing these three components from participants’ NIRS time series in task period. We followed the same approach and removed the first three principal components of the respective rest period of the participants from the NIRS time series of their frontal brain activity that was recorded during the task period. Similar to our NIRS recording, [56] also used 3.0 cm source-detector distance channels. Cooper et al. [57] show that this filter also attenuates the effect of motion artefact (e.g., head motion).

We used the processed time series of the older people’s PFC to calculate their MSE. We used the pattern length m=2, the similarity criterion r=0.15, and the scale factors 1 through 20. We adapted the approach in [58] for computing the MSEs of older people’s frontal brain activity.

### 4.6. Analysis

We computed the individuals’ averaged MSE and applied Friedman test to determine any significance in different media settings. This was followed by post hoc paired Wilcoxon signed-rank test.

We also computed the Spearman correlation between the older people’s averaged MSE of their frontal brain activity and their self-assessed responses to the feeling of fatigue.

The growing adaptation of physically embodied media in elderly care [8,9] along with the similarity of the older people’s pattern of brain activity during in-person and physically embodied communication [6] envision their use for older people’s cognitive training [10]. In fact, through such applications as brain machine interface (BMI) [59], it is foreseeable for these media to act as cognitive training mediators whose exclusive access to older people’s brain activity can help determine whether these individuals have been overcome by their feeling of fatigue, thereby signaling the need for a break in the training session. Therefore, we used the MSE clusters associated with each media setting to determine whether the use of individuals’ averaged MSE can predict their perceived feeling of fatigue during the storytelling. We expected that the change in PFC dynamics to be inversely proportional with such a feeling: the more tired the participants felt, the lower their MSE became. To check the validity of our expectation, we first calculated the mean of the MSE clusters in different media settings and interpreted these mean values as their respective clusters’ decision boundary. Then, we calculated the true positive (TP), true negative (TN), false positive (FP), and false negative (FN) associated with each cluster’s decision boundary. We considered individuals with their averaged MSEs above a given cluster’s decision boundary and their self-assessed responses to the feeling of fatigue ≤4.0 (i.e., on an 8-point scale with 1 = not at all and 8 = very much, Section 4.4) as TP. Similarly, we considered the individuals with their averaged MSEs below the decision boundary and their self-assessed responses to the feeling of fatigue >4.0 as TN. On the other hand, we considered those individuals with their averaged MSEs above the decision boundary while their self-assessed responses to the feeling of fatigue >4.0 as FP and those with their averaged MSEs below the decision boundary while their self-assessed responses to the feeling of fatigue ≤4.0 as FN. We used these values to calculate the accuracy, precision, recall, and F1-score of the MSEs for predicting the older people’s perceived feeling of fatigue. Considering our two-class paradigm (i.e., presence or absence of the feeling of fatigue by older people), the chance level was at 50.0%.

We also report the effect of the physical embodiment on the right-hemispheric PFC in Appendix A.

For Friedman test, we reported the effect size $r=\sqrt{\frac{\chi^{2}}{N}}$ [60] with *N* denoting the sample size. In the case of Wilcoxon test, we used $r=\frac{W}{\sqrt{N}}$ [61] as effect size with *W* denoting the Wilcoxon statistics and *N* the sample size. All results reported are Bonferroni corrected (i.e., multiplying the *p*-values with the sample size, given the use of non-parametric tests). We were unable to collect the self-assessed responses of five participants to the feeling of fatigue and therefore these participants were excluded from correlation and prediction analyses.

We used Python 2.7 for data acquisition and processing. We carried out analyses in Matlab R2016a. We used Gramm [62] for data visualization.

## Figures and Tables

**Figure 1 entropy-21-00199-f001:**
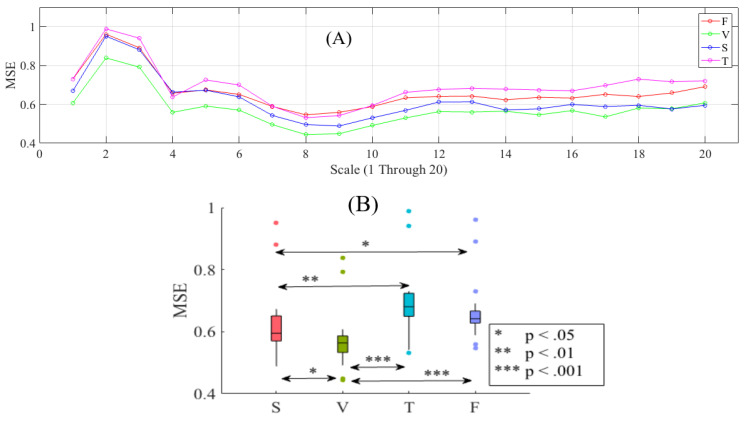
(**A**) Grand-average MSE of older people’s Left-hemispheric PFC activation in speaker (S), video-chat (V), Telenoid (T), and face-to-face (F) settings. In these plots, scale factors 10 and 20 correspond to the one-second and two-second data acquisition intervals, given the sampling rate of our device (i.e., 10.0 Hz). (**B**) Descriptive Statistics of the older people’s left-hemispheric MSE in speaker (S), video-chat (V), Telenoid (T), and face-to-face (F). Asterisks mark the significant differences between these media settings.

**Figure 2 entropy-21-00199-f002:**
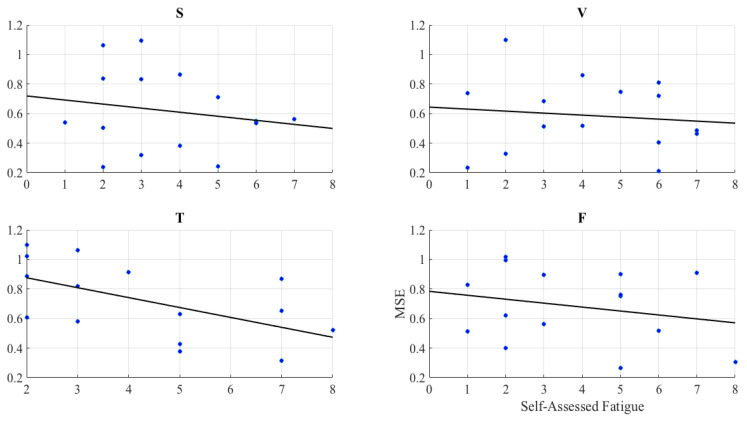
Spearman correlation between MSEs of the older people left-hemispheric PFC activation and their self-assessed responses to feeling of fatigue. (**S**) speaker; (**V**) video-chat; (**T**) Telenoid; (**F**) face-to-face.

**Figure 3 entropy-21-00199-f003:**
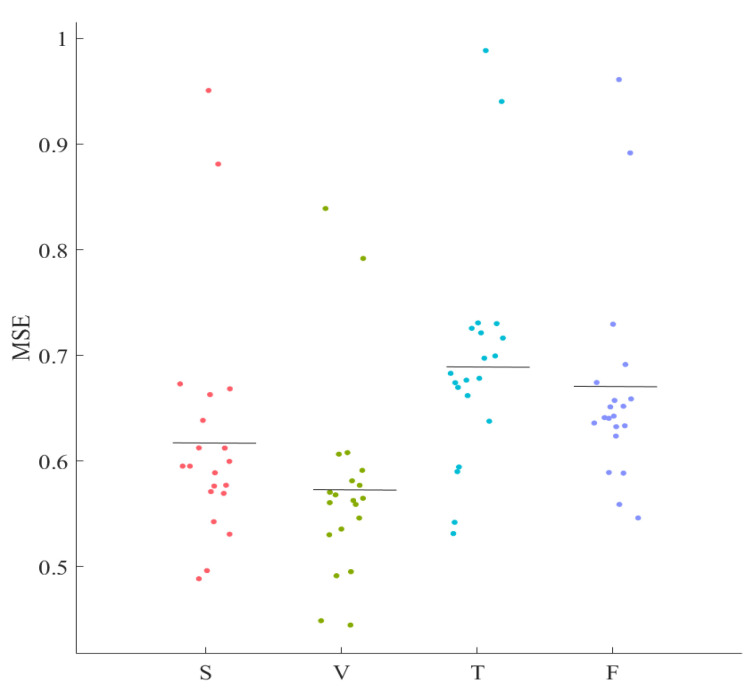
Older people’s left-hemispheric MSE clusters. The boundaries associated with these clusters (i.e., clusters’ mean) are shown (black line segments) in the figure.

**Figure 4 entropy-21-00199-f004:**
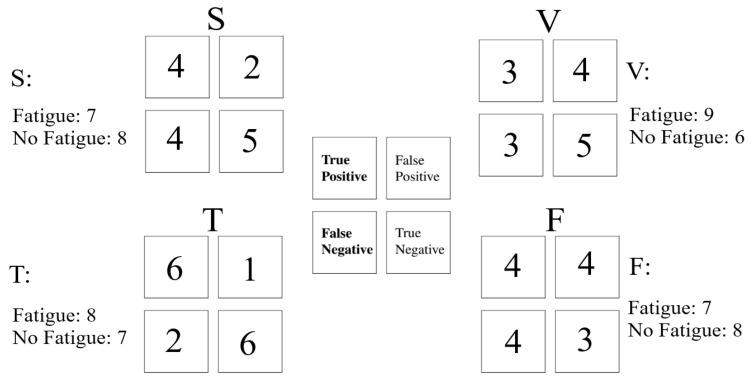
Left-hemispheric MSE vs. self-assessed responses to the perceived feeling of fatigue. In this figure, “Fatigue” and “No Fatigue” refer to the number of older people whose self-assessed responses to the feeling of fatigue was >4.0 and ≤4.0, respectively.

**Figure 5 entropy-21-00199-f005:**
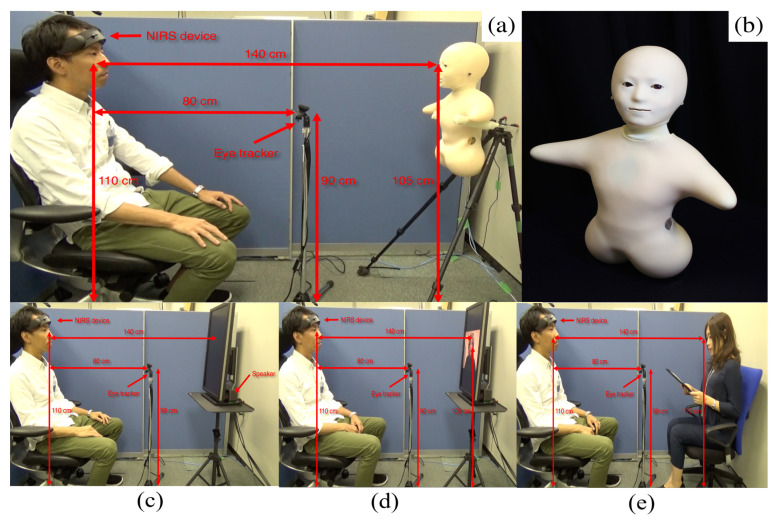
(**a**) Telenoid setting; (**b**) Telenoid medium; (**c**) Speaker setting (**d**) Video-chat setting; and (**e**) face-to-face setting. In these figures, an experimenter demonstrates the experimental setup.

**Figure 6 entropy-21-00199-f006:**
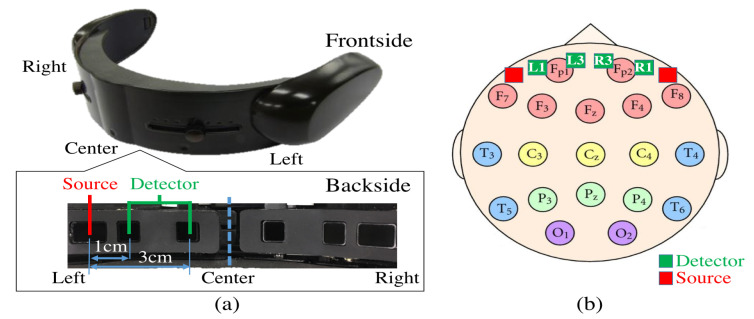
(**a**) fNIRS device in present study. Bottom subplot on left shows arrangement of source-detector of four channels of this device. Distances between short (i.e., 1.0 cm) and long (i.e., 3.0 cm) source and detector of left and right channels are shown. (**b**) Arrangement of 10–20 International Standard System: In this figure, relative locations of channels of fNIRS device in our study (i.e., L1,L3,R1, and R3) are depicted in red (i.e., sources) and green (i.e., detectors) squares. L1,R1,L3, and R3 are channels with short (i.e., 1.0 cm) and long (i.e., 3.0 cm) source-detector distances.

**Table 1 entropy-21-00199-t001:** Left PFC: Prediction accuracy, true positive (TP), true negative (TN), false positive (FP), false negative (FN), and F1-score. Significantly above chance (i.e., 50.00%, given two-class classification) prediction accuracy of the older people’s feeling of fatigue in Telenoid setting is apparent in this table.

Medium	Accuracy	TP	TN	FP	FN	Precision	Recall	F1-Score
S	60.00%	4	5	2	4	0.67	0.50	0.57
V	53.33%	3	5	4	3	0.43	0.50	0.46
T	80.00%	6	6	1	2	0.86	0.65	0.80
F	46.67%	4	3	4	4	0.50	0.50	0.50

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
