# Peer review of "Multiscale Entropy Quantifies the Differential Effect of the Medium Embodiment on Older Adults Prefrontal Cortex during the Story Comprehension: A Comparative Analysis"

_entropy, 2019, doi:10.3390/e21020199_

Round 1

Reviewer 1 Report

This manuscript is interesting, which falls in the scope of Entropy. Following comments should be addressed before the acceptance of publication.

The language should be improved;

The structure of this manuscript is different from a traditional research article, which should be improved.

Author Response

Reviewer 1

We are thankful for the reviewer’s time and kind cooperation to review our manuscript.

Below, we provide our responses to the reviewer’s comment.

Reviewer’s Comment: The language should be improved

Authors’ Response: We have further revised the draft of our manuscript to improve its quality and readability.

Reviewer’s Comment: The structure of this manuscript is different from a traditional research article, which should be improved

Authors’ Response: We apologize for the inconvenience that it may have caused you. For the initial submission, we used the template by the Journal of Entropy. For the current version, we have used the file that we have received from the Journal’s Editorial, as per their advice.

Sincerely,

Soheil

Reviewer 2 Report

The study is carefully designed and the analysis is adequate. The conclusion is supported by the experiment. One concern is about the use of face-to-face vocal input in the experiment. As the authors mentioned, a testing performed face-to-face between persons may incur affect from other factors. Also, is it possible that the fact that the highest accurate result was obtained in telenoid test can be attributed to the curiosity toward the telenoid? After all, it is good that the authors include face-to-face testing in the experiment.

Another concern is about the selection of the targeted responses. The authors did not mention whether there was a particular reason why fatigue response was chosen as a target state in the analysis. Would choosing a certain emotion as the target state be more meaningful? 

In general, the paper is well written, the experimental design is thoroughly examined, and the conclusion is meaningful. 

Author Response

Reviewer 2

We are thankful for the reviewer’s time and kind cooperation to review our manuscript. Below, we provide our responses to the reviewer’s comment.

Reviewer’s Comment: The study is carefully designed and the analysis is adequate. The conclusion is supported by the experiment. One concern is about the use of face-to-face vocal input in the experiment. As the authors mentioned, a testing performed face-to-face between persons may incur affect from other factors.

Authors’ Response: As the reviewer pointed, one concern with regard to the older people’s brain activity was the potential impact of in-person communication (please see references [35-38], [43], [45] from computer-mediated-communication literature (CMC)). However, the observed higher MSEs in the case of in-person and Telenoid were in fact in line with our recent findings (please see reference [6]) in which we used differential entropy of the older peoples’ prefrontal cortex activity: in both cases we observed a higher quantification of the frontal brain activity in response to the physical embodiment (i.e., in-person and Telenoid vs. the speaker and video-chat system in the current results as well as [6]). These results collectively appear to suggest the potential positive effect of the physical embodiment on the brain activity (at least in the case of older people). However, it is certainly crucial to further investigate these findings. We have highlighted a few cases for future consideration in Section 3.1. Limitations and Future Direction, lines 189-208.

Reviewer’s Comment: Also, is it possible that the fact that the highest accurate result was obtained in Telenoid test can be attributed to the curiosity toward the Telenoid? After all, it is good that the authors include face-to-face testing in the experiment.

Authors’ Response: Novelty effect, as adequately noted by the reviewer, might in fact play a role in our result. We have acknowledged this possibility and the necessity for future research to investigate its potential effect through longitudinal studies in 3.1. Limitations and Future Direction, lines 194-199.

Reviewer’s Comment: Another concern is about the selection of the targeted responses. The authors did not mention whether there was a particular reason why fatigue response was chosen as a target state in the analysis. Would choosing a certain emotion as the target state be more meaningful?

Authors’ Response: We apologize for missing this information. We have provided reason for considering the feeling of fatigue in our study in Section 4.6. Analysis, lines 317-323. In addition, we discussed two scenarios in which the ability of a physically embodied medium to detect the feeling of fatigue can benefit the robot-assistive systems (RAS) in Section 3. Discussion, lines 168-176.

Reviewer 3 Report

In this paper the Authors are proposing to utilize the multiscale entropy (MSE) to investigate the effect of physical embodiment on older peoples’ prefrontal cortex (PFC) activity while listening to the stories.

They have provided evidence that physical embodiment induces a significant increase in MSE of the older peoples’ PFC activity and that such a shift in dynamics of their PFC activation significantly reflects their perceived feeling of fatigue

In the proposed method every subject participated in four sessions. They kept the content of the stories intact in these sessions and randomized the order of the media among the participants without changing the order of the narrated stories.

The results are the step toward the use of brain information for quantification of one of the basic component of the human mental state: perceived fatigue due to the verbally communicated contents.

After carefully reading, I find that this paper is extremely interesting, however in order to further improve I would recommend to improve the references on the background. (I suggest: doi: 10.3390/e20020127, doi: 10.1007/978-3-642-32183-2_95, doi: 10.3390/machines6030036, doi: 10.3390/e20100794, doi: 10.3390/e19040176)

Author Response

Reviewer 3

We are thankful for the reviewer’s time and kind cooperation to review our manuscript.

Reviewer’s Comment: After carefully reading, I find that this paper is extremely interesting, however in order to further improve I would recommend to improve the references on the background. (I suggest: doi: 10.3390/e20020127, doi: 10.1007/978-3-642-32183-2_95, doi: 10.3390/machines6030036, doi: 10.3390/e20100794, doi: 10.3390/e19040176)

Authors’ Response: As per reviewer’s recommendation to improve the quality of our submission, we have included the following references in the current version of our manuscript:

[19] Humeau-Heurtier, A. Evaluation of Systems’ Irregularity and Complexity: Sample Entropy, Its Derivatives, and Their Applications across Scales and Disciplines, Entropy, 2018, 20, 794.

[20] Gao, Y., Villecco, F., Li, M. & Song, W., Multi-Scale permutation entropy based on improved LMD and HMM for rolling bearing diagnosis, Entropy, 2017, 19, 178.

[27] Liao, F., Cheing, G.L., Ren, W., Jain, S. & Jan, Y.K., Application of Multiscale Entropy in Assessing Plantar Skin Blood Flow Dynamics in Diabetics with Peripheral Neuropathy, Entropy, 2018, 20, 127.

Please kindly be informed that our choices are solely based on the closeness of the topics of these references to the present manuscript. Otherwise, all these findings are significant contributions to the research community of their respective field.
